# Synergistic and Antibiofilm Effects of the Essential Oil from *Croton conduplicatus* (Euphorbiaceae) against Methicillin-Resistant *Staphylococcus aureus*

**DOI:** 10.3390/ph16010055

**Published:** 2022-12-30

**Authors:** Genil Dantas de Oliveira, Wilma Raianny Vieira da Rocha, José Filipe Bacalhau Rodrigues, Harley da Silva Alves

**Affiliations:** 1Postgraduate Program in Pharmaceuticals Sciences, Department of Pharmacy, State University of Paraiba, Campina Grande 58429-500, Brazil; 2Postgraduate Program in Materials Science and Engineering, Department of Materials Engineering, Federal University of Campina Grande, Campina Grande 58429-900, Brazil

**Keywords:** *Croton* genus, knife breaker, volatile compounds, *Staphylococcus aureus*, resistance profile, antibiofilm

## Abstract

Bacterial resistance refers to the ability of bacteria to resist the action of some antibiotics due to the development of adaptation and resistance mechanisms. It is a serious public health problem, especially for diseases caused by opportunistic bacteria. In this context, the search for new drugs, used alone or in combination, appears as an alternative for the treatment of microbial infections, and natural products, such as essential oils, are important in this process due to their structural diversity, which increases the probability for antimicrobial action. The objective of this study was to extract and identify the chemical components of the essential oil from *Croton conduplicatus* (EOCC), to evaluate the antimicrobial activity, to investigate the effect of the interaction between the EOCC and different antibiotics and to evaluate its antibiofilm potential. The EOCC was obtained by hydrodistillation. Based on chemical characterisation, 70 compounds were identified, with 1.8 cineole (13.15%), p-cymene (10.68%), caryophyllene (9.73%) and spathulenol (6.36%) being the major constituents. The minimum inhibitory concentration (MIC) values of EOCC were 256 and 512 µg mL^−1^ for methicillin-sensitive and -resistant *Staphylococcus aureus* strains (MSSA and MRSA), respectively. The combinations of EOCC with the antibiotics oxacillin and ampicillin were synergistic (OXA/EOCC and AMP/EOCC combined decreased the OXA MIC and AMP MIC to 0.5 and 0.25 for MSSA, respectively, and OXA/EOCC and AMP/EOCC combined decreased the OXA MIC and the AMP MIC to 1 and 0.5 for MRSA, respectively) and could modify the resistance profile of MSSA and MRSA strains. The results indicated that EOCC was also able to partially inhibit biofilm formation. Our study presents important information about the chemical composition of EOCC and its antimicrobial potential and provides a reference to determine the mechanisms of action of EOCC and its use in pharmaceutical formulations.

## 1. Introduction

*Staphylococcus aureus* is a Gram-positive, opportunistic bacterial species characterised by grouped cocci and clusters of cocci that are found mainly in the nasal microbiota. They can cause an infectious condition when they get into the bloodstream by breaking through mucous membrane or skin tissue [1]. This species is highly pathogenic, virulent and shows considerable resistance to environmental factors. A major concern worldwide is the capacity for multi-resistance to antibacterial agents used to combat Gram-positive bacterial infections, such as beta-lactams, glycopeptides and oxazolidones. One of the bacterial resistance profiles of great concern today are those of methicillin-resistant *S. aureus* (MRSA) strains. Generally, MRSA infections are linked with increased difficulty of treatment, morbidity, mortality and high costs to health services [2,3].

The problems related to MRSA strains become more severe when this pathogen develops a biofilm, a structure formed by polymeric substances, proteins and extracellular DNA [4]. This biofilm can form on the surface of medical materials intended for surgical application. Thus, with high resistance to antimicrobials, MRSA strains contaminate patients to the point of making their treatment difficult, increasing life risks, hospital stay length and, consequently, costs. [5]. In view of the complications related to the treatment of *S. aureus* and its resistant strains, The World Health Organization has published a list classifying methicillin-sensitive *S. aureus* (MSSA) and MRSA as high-priority pathogens in the search for new compounds to aid therapy [6].

Due to the increase in bacterial resistance, the development of new bioactive compounds is one of the strategies used in the search for products that may have activity against these pathogens [7,8]. Essential oils (EOs) have emerged as an alternative to combat these microorganisms. Normally, these EOs are a mixture of phenylpropanoids or terpenes (monoterpenes, sesquiterpenes and/or diterpenes), which have different chemical functions, such as alcohols, ketones, aldehydes, and that can present a diverse range of biological activities, including antibacterial activity [9]. 

Several studies evidence the inhibition of MSSA and MRSA strains by EOs and the synergistic relationship between EOs and antibacterials, highlighting their efficiency [3,10,11].

The genus *Croton*, belonging to the family Euphorbiaceae, contains species that produce EO [12]. *Croton conduplicatus* is a native plant species of the Brazilian caatinga and widely used in folk medicine to combat headaches, indigestion and influenza [13]. In a previous study, the essential oil obtained from fresh leaves of *C. conduplicatus* was evaluated by gas chromatography and mass spectrometry (GC-MS), and 42 chemical compounds were identified, of which 1,8-cineole and p-cymene were the major compounds. This EO showed anxiolytic, sedative and antinociceptive activities in an in vivo study using mice [14]. 

Several studies show that 1,8-cineole and p-cymene have antibacterial potential and that can reduce the resistance of MRSA and MSSA strains against antimicrobials [15,16,17,18]. Thus, the objective of this study was to carry out the extraction and chemical characterisation of the EO from the dried leaves of *C. conduplicatus* by GC-MS, to evaluate the antimicrobial activity of the EO from *C. conduplicatus* (EOCC) against strains of *S. aureus* sensitive to methicillin (MSSA), methicillin-resistant *S. aureus* (MRSA) and other microorganisms and to investigate the in vitro interaction of EOCC with conventional antimicrobials against MSSA and MRSA strains and its antibiofilm activity.

## 2. Results

### 2.1. Chemical Characterisation of C. conduplicatus Essential Oil

Essential oils have a high concentration of bioactive compounds such as terpenes, sesquiterpenes, phenolic compounds, phenylpropanoids, non-terpenic aliphatic compounds and heterocyclic compounds, which are responsible for the biological activity of these oils. 

Analysis of the chemical composition of the EOCC was performed by GC-MS. Table 1 shows the chemical composition, the retention indices, the retention time, and the relative percentage of each constituent present in EOCC.

We verified the presence of 74 distinct peaks, as shown in the Figure 1, of which 70 were identified (Table 1), accounting for 95.94% of the chemical composition of EOCC. The monoterpenes 1,8-cineole (13.15%) and p-cymene (10.68%) and the sesquiterpenes caryophyllene (9.73%) and spathulenol (6.36%) were the major compounds (Figure 2). The other compounds with percentages below 5% were considered minor, such as α-pinene (4.93%), bicyclogermacrene (3.40%), α-phellandrene (3.08%), β-pinene (2.77%) and linalool (2.39%).

Among the major compounds 1,8-cineole, p-cymene, caryophyllene and spathulenol were identified by other authors as part of the main constituents of the chemical composition of the EO from *C. conduplicatus* [20,21,22,23,24]. In their results, these compounds showed similar average concentrations to those found here, with the exception of 1,8-cineole, which showed a concentration of 24.09% [21]. In addition, the compounds bicyclogermacrene and α-phellandrene were identified here as minor compounds, whereas in other studies, they were among the major constituents [20,21,22]. Differences in the number and identity of compounds were also identified in the GC-MS analyzes performed by [14] which revealed the presence of 50 peaks and 42 compounds identified in the EOCC obtained from fresh leaves of *C. conduplicatus*. The monoterpenes 1,8-cineol (21.42%) and p-cymene (12.41%) and the sesquiterpenes spathulenol (15.47%) and caryophyllene oxide (12.15%) were considered the majority constituents of the sample (Table 2). In our study, it was possible to identify and quantify 70 different chemical compounds, of which the monoterpenes 1,8 cineole (13.15%) and p-cymene (10.68%) were the majority, as well as the sesquiterpenes caryophyllene (9.73%) and spathulenol (6.36%). This variation in the chemical composition of the EOCC may be related to factors such as the specific location of leaf collection, growing season, botanical origin, climatic factors [25] and the drying process that was used on the plant material.

### 2.2. Antimicrobial Activity of C. conduplicatus Essential Oil

The EOCC showed antibacterial activity, with an minimum inhibitory concentration (MIC) of 256 µg mL^−1^ for the MSSA strain and 512 µg mL^−1^ for the MRSA strain (Figure 3). The bactericidal effect of EOCC was observed only in the presence of twice the MIC concentration. No antimicrobial activity of EOCC was observed against *E. coli*, *P. aeruginosa* and *C. albicans* strains because the MIC values against these strains were > 1024 µg mL^−1^ (Table 3). The MRSA strain used in this study presented a resistance profile to oxacillin (OXA) and ampicillin (AMP), which was detected through the determination of the MIC, highlighting the MIC of OXA of 32 µg mL^−1^, which is used for the detection of methicillin resistance, according to the breakpoints defined by the CLSI document M100 [26]. Because EOCC showed activity against MRSA and MSSA strains, these were selected for the in vitro combination step with antibiotics to investigate the synergistic effect and reduction in MIC of both the antibiotic and EOCC.

### 2.3. Synergistic Activity of C. conduplicatus Essential Oil with Oxacillin and Ampicillin against S. aureus

A synergistic effect of combining subinhibitory concentrations of EOCC (≤ 1/2 MIC) with OXA and AMP was observed against MSSA and MRSA (Figure 4a,b) strains, with MIC reduction percentages ranging from 75% to 96.9%; the synergistic effect was determined by fractional inhibitory concentration (FICi) values that ranged from 0.0938 to 0.3125. Percentage reductions in the MIC of EOCC alone were also observed when combined with OXA and AMP. Thus, the EOCC showed potential to reduce OXA and AMP MIC, and these antibiotics showed potential to reduce the MIC of EOCC (Table 4).

The combinations of EOCC with OXA and AMP were able to reverse resistance to these antibiotics, indicating that at concentrations lower than MIC, OXA and AMP became active against the strains when combined with EOCC.

For the MSSA strains, we verified a reduction in the MIC of both EOCC and OXA and AMP, indicating that EOCC in subinhibitory concentration in combination with these antibiotics has the potential to reduce the MIC for both methicillin-sensitive and methicillin-resistant strains (Table 4). Because of the in vitro combination tests using EOCC with the antibiotics OXA and AMP, it can be stated that these interactions had a synergistic effect.

The results for MSSA and MRSA strains in terms of sensitivity and resistance to methicillin, respectively, are shown in Figure 1. For the MSSA strain, the MIC of isolated OXA was 2 µg mL^−1^. In combination with EOCC, the MIC of OXA was reduced to 0.5 µg mL^−1^, changing its profile from resistant to sensitive to this antibiotic. The reduction in OXA MIC was also observed in the MRSA strain (Figure 5B). A similar effect was observed in the associations of EOCC with AMP, with a synergistic effect against the MRSA strain, with a reduction in the MIC of AMP from 16 to 0.5 µg mL^−1^ (Figure 5D).

### 2.4. Antibiofilm Activities of C. conduplicatus Essential Oil

The antibiofilm activity of EOCC alone (at an concentration equal to the MIC) and in combination with OXA and AMP was measured against MRSA and MSSA strains. The percentages of biofilm inhibition reducing formed biofilms were evaluated (Figure 6). Figure 7 shows the antibiofilm activity of OECC and the combinations with OXA and AMP against the MRSA strain.

The isolated EOCC at MIC showed inhibited biofilm formation in MSSA and MRSA strains by 18% and 22%, respectively. In the evaluation of the ability of EOCC to reduce formed (mature) biofilms of these strains, isolated EOCC was able to reduce the biofilm of the MSSA strain by 32% and that of the MRSA strain by 27%, highlighting the inhibition or reduction of biofilms by EOCC.

The EOCC combined with the antibiotics OXA and AMP, in subinhibitory concentrations, also showed activity, for most of the combinations, against the tested strains, with emphasis on the reduction of mature biofilm by the combination EOCC/OXA that was statistically equal (*p* > 0.05) the inhibition caused by EOCC (MIC concentration) against the strain MSSA and, the combination EOCC/OXA was also able to reduce the biofilm formed by the strain MRSA.

An inhibition of biofilm formation by EOCC and its combinations with OXA and AMP at concentrations lower than the MIC was also observed against the *S. aureus* strains tested, which was more pronounced for the combination EOCC/AMP against the MRSA strain.

The treatments applied, both regarding the inhibition of biofilm formation and of mature biofilm, showed differences only to the positive control used, indicating that the treatments were effective against MSSA and MRSA antibiofilm activities. Although there was no significant difference between the treatments, the results indicate that EOCC combined with OXA or AMP at subinhibitory concentrations has an effect similar to that of EOCC alone at a higher concentration (equivalent to the MIC).

## 3. Discussion

The chemical composition of the essential oil from *C. conduplicatus* leaves was obtained via GC-MS. Based on the results, it is rich in bioactive compounds, mainly monoterpenes and sesquiterpenes. Generally, EOs are natural products rich in bioactive compounds, such as monoterpenes and phenylpropanoids. These compounds are used by plants as a defense against predators and microorganisms, including those that are pathogenic to humans [27]. 

Studies indicate that the EOs from plants belonging to the genus *Croton* have potential antibacterial activity. For example, Barbosa [28] reported that the antimicrobial activity of *C. urticifolius* EOs against strains of *S. aureus* and *E. coli*, and Rocha [29] observed that EOs obtained from the leaves of *C. tetradenius* and *C. pulegiodorus* inhibited the growth of clinical isolates of *S. aureus*, leading to cell death.

Other EOs from plants of the genus *Croton* with a chemical composition similar to that of *C. conduplicatus* also possess antimicrobial activity against clinically important strains, such as the activity of *C. heliotropiifolius* [21], *C. ferrugineus* [30], *C. adipatus*, *C. thurifer* and *C. collinus* [31] essential oils against *S. aureus, Klebsiella pneunomiae*, *Enterococcus faecalis, Candida albicans, Mycobacterium tuberculosis, Escherichia coli* and *Pseudomonas aeruginosa.* The activity of *C. cajucara* EO against an MRSA strain was related by Azevedo et al. [32], who verified that the EO of this species contained 7-hydroxy-calamenene as the major component, with an MIC value of 4.760 μg mL^−1^.

The antimicrobial properties are related to the bioactivity of the major compounds as well as to their synergistic action with the minor EO constituents [16,33]. This study presents the first report of the anti-MRSA activity of *C. conduplicatus* EO.

Studies have associated the antimicrobial action of EOs to the high concentration of 1,8-cineole [15,16], with activity against *S. aureus*, *Escherichia coli*, *Micrococcus luteus* and *Bacillus subtilis*, among other microorganisms [16,34].

The monoterpene 1,8-cineole, the major component of EOCC, inhibits various microbial strains [35,36,37] including MRSA strains [38]. It shows antibiotic activity both toward the bacteria, such as *Escherichia coli*, *Klebsiella pneumoniae* and *Pseudomonas aeruginosa*, and the biofilms of pathogenic yeasts, such as *Candida albicans* [39,40,41,42].

The antibacterial activity of 1,8-cineole is associated with oxidative stress and damage to the bacterial cell membrane, causing extravasation of the intracellular contents [36]. Previous studies also reported its synergistic and isolated activity, with a consequent reduction in the MIC of the antibiotic mupirocin and betalactamic antibiotics, such as penicillin, respectively, against MRSA strains [38]. A previous study [3] proved the anti-biofilm and anti-quorum-sensing activity toward MRSA strains, highlighting the importance of combating microbial biofilms to avoid complicated infections and the spread of these strains in hospitals.

Other studies also reported the action of 1,8-cineole, p-cymene, caryophyllene and spathulenol against a broad spectrum of microorganisms, including multidrug-resistant bacteria [17,18,43,44,45]. One of the proposed mechanisms indicates that these compounds permeate the cell wall of bacteria, reversing their resistance and resensitising them to antibiotics [18,45]. A previous study showed the antibacterial activity of EO containing p-cymene against MRSA strains [46]. This compound presents a greater inhibitory potential when associated with other monoterpenes, such as carvacrol and 4-terpineol [47,48]. The p-cymene can affect the membrane integrity of MSSA and MRSA strains, facilitating the passage of other antimicrobial agents and modifying the resistance of these strains to certain antibacterials [49]. Its antibiofilm activity was reported by Miladi et al. [50], who found that p-cymene alone and in combination with tetracycline was effective in preventing biofilm formation in MRSA and MSSA strains as well as clinical *S. aureus* strains isolated from the human oral cavity.

However, no studies addressed the isolated action of caryophyllene and spathulenol in inhibiting MRSA and MSSA strains. However, EO that contained caryophyllene or spathulenol as one of its major compounds inhibited the action of these strains and their biofilms [51,52,53,54]. This points to the development of studies investigating the potential activity of these compounds in inhibiting MRSA or MSSA strains.

Thus, the synergistic activity of EOCC in combination with betalactamic antibiotics, such as OXA and AMP used in this study, may be related to the joining of the mechanisms of action. The EOCC, containing 1,8-cineole, p-cymene, caryophyllene and spathulenol as major components, causes damage to the cell membrane, with extravasation of intracellular contents, and betalactamic antibiotics act by inhibiting penicillin-binding proteins (PBP), preventing cell wall formation [55]. The combination of the mechanisms of action potentiates both the action of EOCC and those of OXA and AMP against MSSA and MRSA strains.

The MRSA strains have a genetic mutation that results in the production of an alternative PBP, namely PBP2, with a low affinity for penicillin [55], thus ensuring broad resistance to betalactams, except ceftaroline and ceftobiprole [56]. The presence of PBP2 in the cell wall of *S. aureus* is an example of a specific microbial resistance. Microbial biofilm formation, on the other hand, is a virulence factor that causes nonspecific resistance, especially when prostheses, catheters and other invasive medical devices are infected, in addition to the relationship with endocarditis and osteomyelitis [57].

Given the evidence regarding the bioactivity of the major compounds present in EOCC, it could be inferred that EO from *C. conduplicatus* presents a potential antimicrobial effect, especially against MRSA strains. This study considerably contributes to the knowledge about plants of the genus *Croton*, especially regarding the species *C. conduplicatus*; we confirm the antibacterial activity, synergistic activity against MRSA and MSSA strains and antibiofilm activity of the EO from the leaves of this plant species.

## 4. Materials and Methods

### 4.1. Identification and Harvesting of Plant Material

Leaves from *C. conduplicatus* were collected at Irecê City, Bahia, at −11.345982 S, −41.891216 W, at 14:00 h on 4 February 2021. For the proper botanical identification of the collected plant material, an exsiccate was made and later deposited and registered under number 3217 in the Manoel de Arruda Câmara Herbarium (ACAM) of the State University of Paraiba.

### 4.2. Obtaining Plant Drug

The leaves from *C. conduplicatus* were separated from the other aerial parts and then dried in a circulating oven at a temperature of 40 °C for a period of 72 h. The dried leaves were ground in a knife mill to a particle size of approximately 10 mesh, and 704.5 g of the plant drug (PD) was obtained and stored in a hermetically sealed container protected from light.

### 4.3. Essential Oil Extraction

The EO was obtained through the hydrodistillation technique at 100 °C, using the simple Clevenger apparatus and a heating mantle (Warmnest, UK), for 3 h. The entire PD was used, considering a ratio of 700 mL of distilled water for every 100 g of PD, the amount of distilled water needed to cover it completely. This procedure enabled the extraction of 2.2 mL of *C. conduplicatus* essential oil (EOCC), as shown in Figure 8. The obtained EOCC was stored under refrigeration.

### 4.4. Gas Chromatography Coupled to Mass Spectrometry (GC-MS)

Gas chromatography-mass spectrometry (GC-MS) analysis was conducted on a Clarus 680 gas chromatograph equipped with a PALCOMBI-xt automatic injector, an Elite-5MS column (30 m × 0.25 mm i.d., 0.25 μm) and a Clarus SQ8S mass spectrometer (Perkin Elmer, Waltham, MA, USA). Helium gas was used as a carrier gas at a flow rate of 1 mL/min. The injector was heated to 250 °C with a 1:10 split, and 1.0 μL of the sample was injected. The oven was programmed as follows: 1st step: heating gradient at 35 °C (for 2 min) to 90 °C (for 2 min) at a rate of 10 °C/min; 2nd step: 90 to 130 °C (for 4 min) at a rate of 8 °C/min; 3rd step: 130 to 230 °C (for 2 min) at a rate of 4 °C p/min. The analysis time was 45.50 min. The detector worked in electron ionisation (EI) mode at 70 eV, with an interface temperature of 180 °C (inlet line) and a source (source temp) at 220 °C. Mass fragments were monitored in the range of 40–610 Da. The NIST database from the NIST MS Search Version 2.2 software (National Institute of Standards and Technology, Gaithersburg, MD, USA) was used for identification of the compounds.

The compounds present in the EOCC were identified by comparing their respective mass spectra with those of other previously analysed compounds and with the mass spectra of the NIST database (NIST MS Search Version 2.2), with the retention index (RI) of each compound and by comparing the chemical composition of *C. conduplicatus* essential oils described in other studies [14,20,21,22]. These analyses were carried out in triplicate.

### 4.5. Antimicrobial Activity of C. conduplicatus Essential Oil

#### 4.5.1. Microbial Strains and Inoculum Standardisation

The microbial strains selected for this study were obtained from the American Type Culture Collection (ATCC): Methicillin-sensitive *Staphylococcus aureus* (MSSA) ATCC 25923, methicillin-resistant *Staphylococcus aureus* (MRSA) ATCC 33591, *Escherichia coli* ATCC 25922, *Pseudomonas aeruginosa* ATCC 27853 and *Candida albicans* ATCC 10231; the strains were kept in the Laboratory of Clinical Analysis of the State University of Paraíba (LAC-UEPB), stored in brain heart infusion broth (BHIB) (DIFCO^®^) and 20% (*v/v*) glycerol (LGCBIO).

The inoculum of *S. aureus*, *E. coli* and *P. aeruginosa* strains was standardised according to the document M07 of the Clinical Laboratory Standards Institute-CLSI [58], starting from a 24-h culture in Mueller Hinton broth (MHB) (DIFCO^®^). The inoculum of *C. albicans* was standardised according to the CLSI document M27 [59] from a 24-h culture with isolated colonies on potato dextrose agar (PDA). The initial inoculum was standardised to reach a concentration equal to MacFarland’s 0.5 scale in MHB for bacteria and in Sabouraud broth (SB) (DIFCO^®^) for yeast.

For the antimicrobial screening assay, the initial inocula were diluted to obtain a concentration of 2.0 to 8.0 × 10^5^ CFU/mL for the bacteria and 1.0 to 5.0 × 10^3^ CFU/mL for *C. albicans.*

#### 4.5.2. Antimicrobial Agents

Oxacillin sodium (OXA) and ampicillin sodium (AMP), obtained from Teuto Laboratory, were used as antimicrobial agents. Polymyxin B sulphate was obtained from Eurofarma and Amphotericin B obtained from Cristalia Laboratory.

For sample preparation, the EOCC was solubilised in a mixture of 3 mL of dimethylsulfoxide (Neon), 2 mL of Tween 80^®^ (Dinâmica) and 6 mL of sterile deionised water. The antimicrobials were solubilised in sterile distilled water. All solutions were filtered through a 0.22-µm membrane before activity testing to ensure sterility.

#### 4.5.3. Antimicrobial Screening

Antimicrobial screening was performed by determining the minimum inhibitory concentration (MIC) and the minimum bactericidal and fungicidal concentration (MBC/FMC).

The MIC of EOCC and antimicrobials was determined by the broth microdilution method as described in the CLSI documents M07 [58] and M27 [59]. For this, serial dilutions of these compounds were performed to obtain final plaque concentrations ranging from 1024 to 0.03 µg/mL. Subsequently, 100 µL of each of these dilutions was added to a 96-well plate and received 100 µL of the standardised inoculum in each well to obtain a final concentration of 1.0 to 4.0 × 10^5^ CFU/mL for the bacterial strains and of 0.5 to 2.5 × 10^3^ CFU/mL for *C. albicans*.

Growth inhibition was analysed by evaluating growth in broth compared to an untreated growth control by adding a 0.01% resazurin (Sigma-Aldrich^®^, St. Louis, MO, USA) solution. The MIC was taken as the lowest concentration capable of inhibiting growth after 24 h of incubation at 35 °C. A control well of DMSO/Tween 80^®^/water diluent (3.0/1.0/6.0) was included to rule out diluent activity. Wells showing the MIC had a sample seeded onto MHB or PDB plates, which were incubated for 24 and 48 h at 35 °C, respectively, and surviving colonies were counted. The MBC or MFC was considered the lowest concentration capable of inhibiting 99.9% of microbial growth after the incubation period.

#### 4.5.4. Checkerboard Assay against *S. aureus*

Dilutions of EOCC, oxacillin (OXA) and ampicillin (AMP) were performed in MHB. From these dilutions, 50-µL aliquots were added into 96-well microplates to obtain a final concentration equal to eight dilutions lower than the MIC of EOCC and nine dilutions lower than that of OXA and AMP. Then, 100 µL of the standardised suspension of the MSSA and MRSA strains (10^5^ CFU/well) was added to each well. The plates were incubated for 24 h at 35 °C, and growth inhibition was assessed by comparison with the growth control group. The data were interpreted after calculating the Fractional Inhibition Concentration (FICi) values using the following equation:FICi = MIC_EOCC+AA_/MIC_EOCC_ + MIC_EOCC+AA_/MIC_AA_(1)
where: 

MIC_EOCC_: MIC of the essential oil from *C. conduplicatus*;

MIC_AA_: MIC of the antimicrobial agent;

MIC_EOCC+AA_: MIC of EOCC in combination with antimicrobial agents.

The combination was considered synergistic when FICi ≤ 0.5, additive when 0.5 < FICi ≤ 1, indifferent when 1 < FICi ≤ 2, antagonistic when FICi ≥ 2 [60]. Tests were performed in a triplicate of independent experiments.

#### 4.5.5. Activity against *S. aureus* Biofilm Formation

To evaluate the activity of EOCC and its combinations with antifungals at subinhibitory concentrations on the formation of biofilms of *S. aureus*, the methodology described by Manoharan et al. [61], with modifications, was used. Starting from 24-h cultures of MSSA and MRSA strains in MHB, a 50-µL aliquot was inoculated into MHB and then incubated at 35 °C until turbidity equivalent to 0.5 McFarland scale (1.0 × 10^8^ to 2.0 × 10^8^ CFU/mL) was reached. From this culture, a 100-µL aliquot was dispensed into the wells of the microdilution plates. Subsequently, 100 µL of the compounds and their combinations was dispensed at final plate concentrations corresponding to the MIC and the FICi determined via the checkerboard method, and the plates were incubated at 35 ± 2 °C for 24 h. A positive control of biofilm formation was included, containing 100 µL MHB and 100 µL of the inoculum, and the negative control consisted of 200 µL MHB.

The biomass of the biofilm formed after the treatments was determined according to Munusamy, Vadivelu and Tay [23] via staining with crystal violet, with modifications.

After the incubation period, the plates were washed three times with sterile saline solution (NaCl 0.85%) to remove any cells that were not adhered to the biofilm and placed in a drying oven at 40 °C for 20 min. Subsequently, 200 µL of a 0.4% crystal violet solution was added to all wells, and the plates were incubated for 45 min. After incubation, the plates were washed with sterile distilled water and oven-dried at 40 °C for 20 min. After drying the wells, 200 µL of 96% ethyl alcohol was added for 45 min to detain the biofilm.

To perform the reading, 100 µL was removed and placed in a new 96-well flat-bottom plate. Reading was performed by optical density (OD) in a microplate reader (xMark™, Bio-Rad, Hercules, CA, USA) at a wavelength of 585 nm. The results were expressed as the percentage of inhibition of biofilm formation compared to the positive control, which represents 100% of biofilm formation.

#### 4.5.6. Activity against *S. aureus* Formed Biofilm

Starting from a 24-h culture of *S. aureus* strains in MHB, a 200-µL aliquot was inoculated into 20 mL of MHB and incubated at 35 °C until turbidity equivalent to McFarland’s 0.5 scale (1.0 × 10^6^ to 5.0 × 10^6^ CFU/mL) was reached. From this culture, 200 µL was dispensed into all 96 wells of the microdilution plates and incubated at 35 ± 2 °C for 24 h to form the biofilm. After the incubation period, the culture medium was carefully removed, and the wells were aseptically washed three times with sterile saline solution (NaCl) at 0.85% to remove the cells that were not adhered to the wells; subsequently, the plates were sealed and placed on a flat surface for 20 min at room temperature (25 °C) for drying. The dried biofilms were spiked with 200 µL of the compounds and their combinations, and the plates were re-incubated at 35 ± 2 °C for 24 h. A positive control for biofilm formation was included, consisting of 100 µL of MHB and 100 µL of the inoculum, and the negative control consisted of 200 µL of MHB.

To evaluate the activity of EOCC and its combinations with OXA and AMP on biofilms formed by *S. aureus* strains, the methodology described by Uppuluri et al. [62] was used, with modifications.

After the incubation period, the culture medium was removed, and the plates were washed three times with sterile 0.85% NaCl (Dinâmica) to remove any non-adhered cells; subsequently, the plates were placed in an oven at 40 °C for 20 min for drying. The biofilm biomass was determined according to Munusamy, Vadivelu and Tay [23], with some modifications, as described above. The results were expressed as percentage of inhibition of the biofilm formed compared to the positive control (100% of biofilm formation), indicating action against the biofilm formed by *S. aureus*.

#### 4.5.7. Statistical Analysis

The experiments were performed in triplicates of independent tests, and results were expressed as the mean and standard deviation of the percentage of biofilm inhibition, calculated in the Office Excel 2019 software. The results were submitted to statistical analysis by applying the t-test, performing analysis of variance and evaluating the statistical difference among the treatments. Statistical significance was set at *p* < 0.05.

## 5. Conclusions

The EOCC showed antibacterial activity against methicillin-susceptible and methicillin-resistant *Staphylococcus aureus* strains. This effect was observed together with a potential synergistic effect when EOCC was associated with OXA and AMP, to the point of modifying the resistance profile of MSSA and MRSA strains. Furthermore, EOCC also showed a potential effect of inhibiting biofilm formation and reducing mature biofilms of MRSA and MSSA strains. These results were associated with the chemical composition of EOCC, which showed 1,8-cineole, p-cymene, caryophyllene and spathulenol as the main constituents, compounds known for their activity against multidrug-resistant bacterial strains. These results provide a solid reference for further studies on EOCC in combination with different antibiotics to evaluate its mechanism of action and propose pharmaceutical formulations, with the aim to broaden the therapeutic resources against infections caused by pathogenic microorganisms.

## Figures and Tables

**Figure 1 pharmaceuticals-16-00055-f001:**
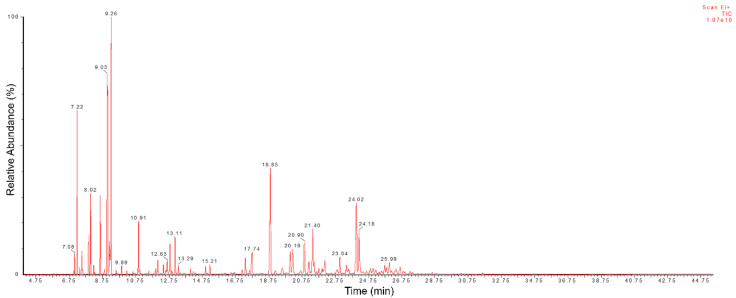
GC-MS total ion chromatogram of *C. conduplicatus* essential oil.

**Figure 2 pharmaceuticals-16-00055-f002:**
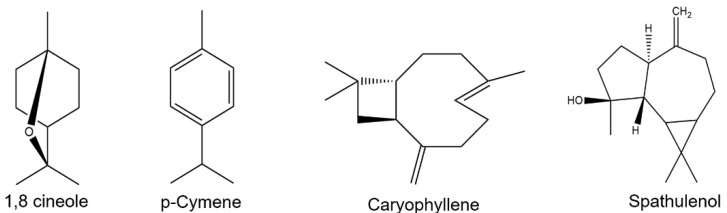
Chemical structures of the major compounds identified in *C. conduplicatus* essential oil via GC-MS analysis.

**Figure 3 pharmaceuticals-16-00055-f003:**
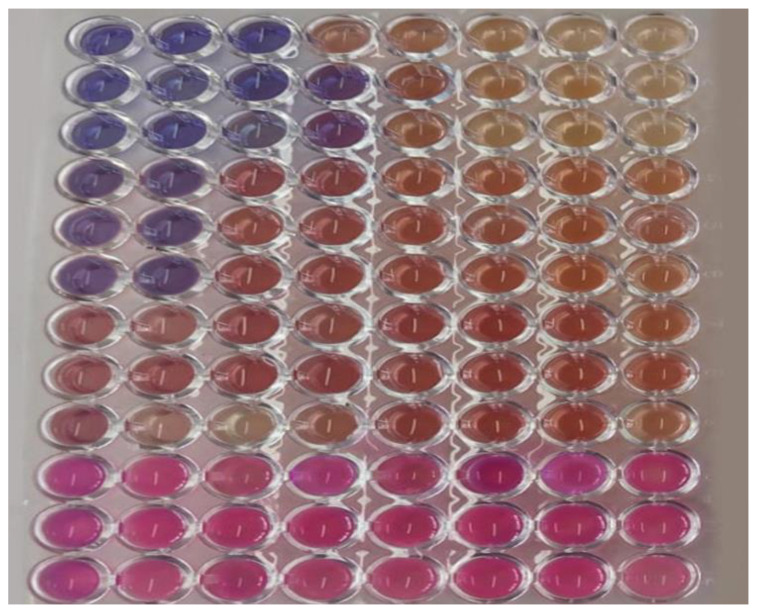
Antimicrobial activity of *C. conduplicatus* essential oil relieved by the addition of 0.01% resazurin solution. Each set of three lines on the plate represents a tested microorganism, with the following strains: methicillin-sensitive *S. aureus* (MSSA), methicillin-resistant *S. aureus* (MRSA), *Escherichia coli* and *Pseudomonas aeruginosa*, respectively.

**Figure 4 pharmaceuticals-16-00055-f004:**
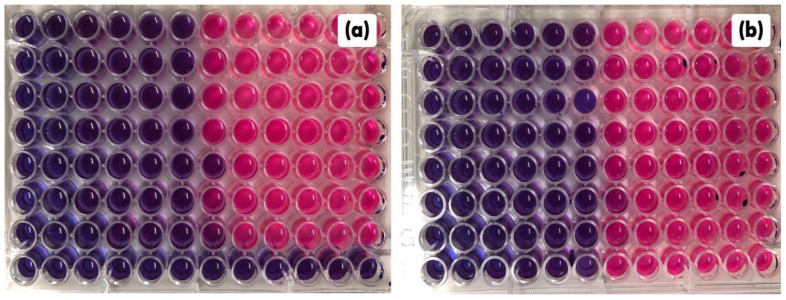
Checkerboard assay between *C. conduplicatus* essential oil with oxacillin (**a**) and *C. conduplicatus* essential oil with ampicillin (**b**) against the MRSA strain. The blue colour indicates the activity of the combined concentration.

**Figure 5 pharmaceuticals-16-00055-f005:**
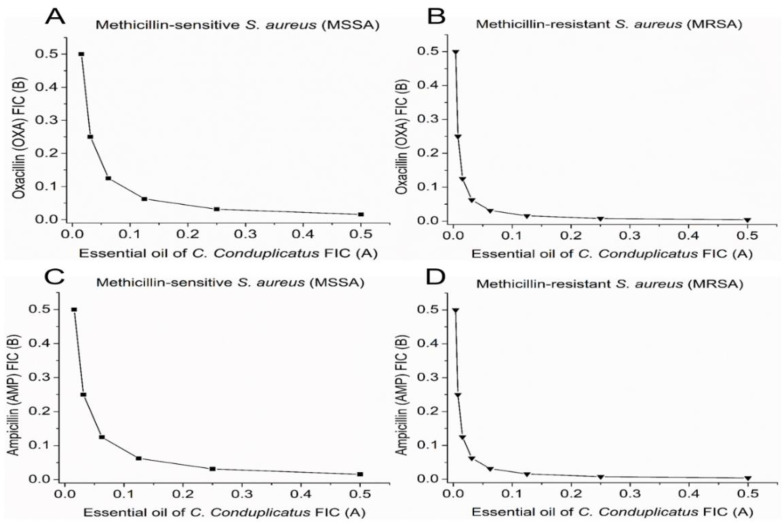
Isobole curves showing the synergistic effect of essential oil from *C. conduplicatus* leaves with (**A**,**B**) oxacillin and (**C**,**D**) ampicillin against methicillin-sensitive *Staphylococcus aureus* (MSSA) ATCC 25923 and methicillin-resistant *Staphylococcus aureus* (MRSA) ATCC 33591, respectively.

**Figure 6 pharmaceuticals-16-00055-f006:**
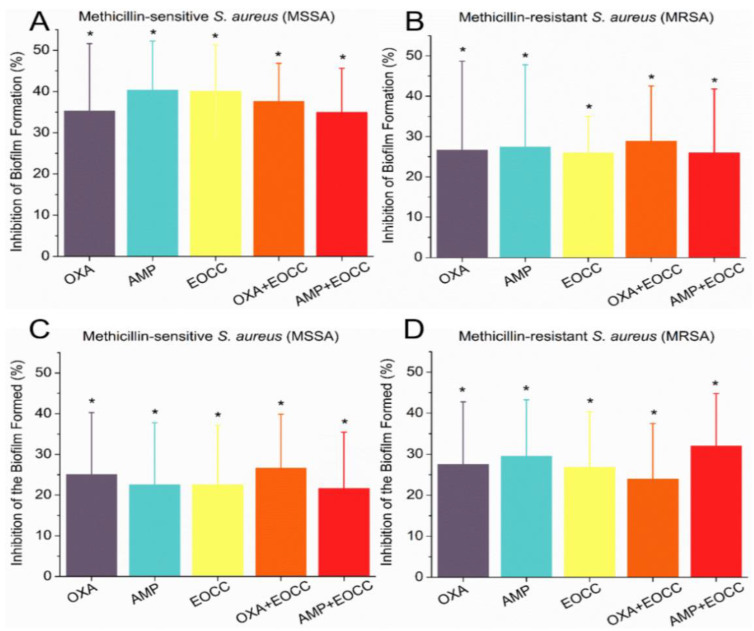
Percentage inhibition of biofilm formation by EOCC, OXA and AMP alone and in combination against the MSSA strain (**A**); percentage inhibition of biofilm formation by EOCC, OXA and AMP in combination and alone against the MRSA strain (**B**); percentage inhibition of mature biofilm for EOCC, OXA and AMP alone and in combination against the MSSA strain (**C**); percentage inhibition of mature biofilm for EOCC, OXA and AMP alone and in combination against the MRSA strain (**D**). The asterisk indicates a non-significant difference (*p* > 0.05) according to the t-test.

**Figure 7 pharmaceuticals-16-00055-f007:**
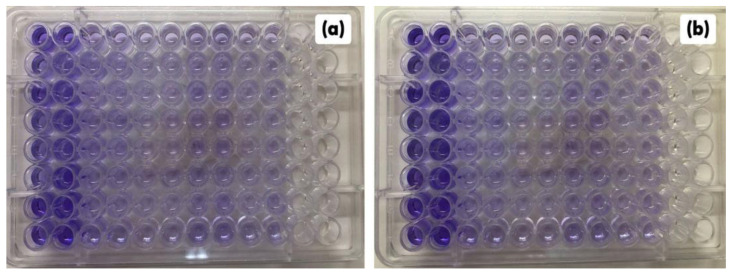
Biofilm biomass stained with violet crystal after experiments of inhibition of biofilm formation (**a**) and activity against mature biofilm (**b**) for EOCC, OXA and AMP alone and in combination against the MRSA strain. The first two columns (1 and 2) in (**a**,**b**) represent the biomass of untreated biofilm; the columns 3 to 10 are the treatments with EOCC and the combinations with OXA and AMP.

**Figure 8 pharmaceuticals-16-00055-f008:**
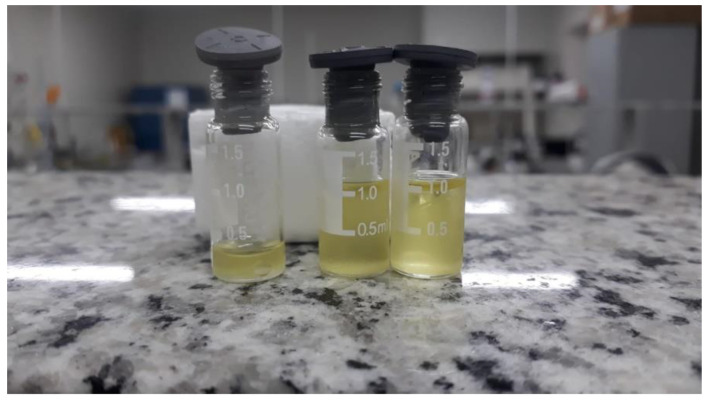
Production of the EOCC using the hydrodistillation technique.

**Table 1 pharmaceuticals-16-00055-t001:** Chemical constituents of *Croton conduplicatus* essential oil.

No	Compounds ^a^	RILit ^b^	RICalc ^c^	RT (min)	Area (%)	No	Compounds ^a^	RILit ^b^	RICalc ^c^	RT (min)	Area (%)
1	α-Tricyclene	926	952	7.02	0.09	37	α-Cubebene	1348	1340	17.33	1.19
2	α-Thujene	930	955	7.08	0.57	38	β-Bourbonene	1388	1350	17.61	0.20
3	α-Pinene	939	960	7.22	4.93	39	β–Elemene	1390	1355	17.74	1.92
4	1-Ethylbutyl Hydroperoxide ^d^	-	966	7.38	0.21	40	Caryophyllene	1419	1397	18.85	9.73
5	Camphene	954	970	7.50	0.73	41	β-Copaene	1432	1409	19.16	0.31
6	Sabinene	975	986	7.92	1.25	42	cis-Eudesma-6,11-diene	1477	1424	19.56	0.62
7	β-Pinene	979	990	8.02	2.77	43	α-Caryophyllene	1454	1442	20.05	1.77
8	β-Myrcene	990	997	8.21	0.33	44	Alloaromadedrene	1460	1448	20.20	1.99
9	2,3-dihydro-1,8-cineole	991	999	8.27	0.05	45	γ-Gurjunene	1477	1466	20.68	0.27
10	α-Phellandrene	1002	1012	8.60	3.08	46	Germacrene D	1485	1474	20.90	2.64
11	α-Terpinene	1017	1021	8.85	0.20	47	β-Selinene	1490	1484	21.17	0.87
12	p-Cymene	1024	1028	9.04	10.68	48	Bicyclogermacrene	1500	1493	21.40	3.40
13	D-Limonene	1029	1033	9.16	1.51	49	α-Muurolene	1500	1496	21.48	0.72
14	β-Thujene ^d^	-	1034	9.20	0.69	50	Eremophila-1(10),8,11-triene ^d^	-	1502	21.65	0.16
15	1,8-Cineole	1031	1037	9.26	13.15	51	Germacrene A	1509	1507	21.77	0.42
16	β-cis-Ocimene	1037	1048	9.55	0.18	52	γ-Cadiene	1513	1513	21.95	0.43
17	γ-Terpinene	1059	1060	9.89	0.40	53	δ-Cadinene ^d^	-	1520	22.13	1.04
18	Cis-Sabinene hydrate	1070	1073	10.22	0.18	54	β-Calacorene	1545	1548	22.86	0.48
19	Isoterpinolene	1088	1087	10.59	0.16	55	Ciclohexane,1,3-diisopropenyl-6-methyl ^d^	-	1554	23.04	1.34
20	Linalool	1096	1099	10.91	2.39	56	Germacrene B	1561	1569	23.43	0.63
21	cis-4-Thujanol	1098	1101	10.98	0.18	57	Cis-α-Copaene-8-ol ^d^	-	1574	23.56	0.51
22	Cis-p-Menth-2-en-1-ol	1121	1122	11.54	0.17	58	Spathulenol	1578	1591	24.02	6.36
23	Trans-pinocarveol	1139	1139	11.98	0.33	59	Ledol	1602	1622	24.85	0.50
24	(+)-Camphor	1146	1142	12.07	0.75	60	Humulene epoxide II	1608	1629	25.02	0.43
25	Pinocarvone	1164	1155	12.41	0.45	61	β-Guayene ^d^	-	1635	25.19	0.44
26	Terpineol <cis-dihydro-a->	1164	1161	12.57	0.23	62	γ-Maaliene ^d^	-	1643	25.39	0.12
27	Borneol	1165	1163	12.63	0.67	63	Epicubebol ^d^	-	1648	25.53	0.26
28	Terpinen-4-ol	1177	1170	12.81	1.44	64	β-Spathulenol ^d^	1578	1655	25.73	0.60
29	p-Cymen-8-ol	1182	1174	12.92	0.19	65	Bicyclo[7.2.0]undecan-3-ol, 11,11-dimethyl-4,8-bis(methylene)- ^d^	-	1660	25.85	0.34
30	α-Terpineol	1188	1181	13.11	1.96	66	10-epi-α –Cadinol	1640	1665	25.99	0.88
31	Cis-sabinol ^d^	-	1188	13.29	0.40	67	α-Muurolol	1646	1670	26.13	0.15
32	Cis-piperitol	1196	1192	13.40	0.09	68	Epi-α-Muurolol	1642	1681	26.40	0.72
33	β-Sabinyl Acetate ^d^	-	1216	14.04	0.22	69	Xantoxyline	1668	1690	26.65	0.65
34	Bornyl acetate	1288	1250	14.94	0.36	70	(1R,7S,E)-7-isopropyl-4,10-dimethylene-cyclodec-5-enol	1686	1713	27.26	0.14
35	Thymol	1290	1260	15.21	0.53	71	NI ^e^	-	1762	28.58	0.14
36	α-Longipinene	1352	1333	17.15	0.34	Total identified					95.94

^a^ Constituents listed in order of elution on Elite-5MS column; ^b^ RILit. = Literature retention index [19]; ^c^ RICalc. = calculated retention index; ^d^ Compounds identified by NIST; ^e^ Not identified.

**Table 2 pharmaceuticals-16-00055-t002:** Comparison of chemical compounds obtained by GC-MS from the essential oil of fresh. leaves of *C. conduplicatus* (Oliveira Júnior et al., 2018).

Peak	Compounds	RT (min)	% GC-MS
1	Tricyclene	8.447	0.08
2	α-Thujene	8.726	0.50
3	α-Pinene	8.927	2.30
4	Camphene	9.465	0.49
5	Sabinene	10.521	1.46
6	α-Phellandrene	11.731	1.44
7	*p*-Cymene	12.574	12.41
8	1,8-Cineole	12.792	21.42
9	NI	13.694	0.07
10	γ-Terpinene	13.942	0.14
11	Terpinolene	15.087	0.05
12	(*E*)-Sabinene	15.569	0.03
13	NI	15.716	0.13
14	(*Z*)-*p*-Menth-2-en-1-ol	16.402	0.16
15	α-Campholenal	16.535	0.01
16	(*E*)-Pinocarveol	16.977	0.18
17	Camphor	17.117	0.32
18	Pinocarvone	17.842	0.09
19	Borneol	17.989	0.52
20	NI	18.130	0.05
21	Terpinen-4-ol	18.417	2.28
22	α-Terpineol	19.001	0.60
23	Isobornyl acetate	22.236	0.32
24	α-Copaene	25.167	0.20
25	β-Bourbonene	25.450	0.21
26	β-Elemene	25.713	0.34
27	(*E*)-Caryophylene	26.560	7.52
28	α-Humulene	27.613	1.55
29	Alloaromadendrene	27.841	1.69
30	Germacrene D	28.473	0.31
31	β-Selinene	28.628	0.32
32	Bicyclogermacrene	28.955	1.61
33	δ-Amorphene	29.488	0.58
34	δ-Cadinene	29.776	0.53
35	α-Calacorene	30.355	0.14
36	NI	30.611	0.32
37	Spathulenol	34.413	15.47
38	Caryophyllene oxide	31.541	12.15
39	Ledol	32.105	1.50
40	Humulene epoxide	32.265	1.42
41	Cubenol	32.450	0.20
42	Acorenol	32.826	0.25
43	NI	32.966	0.44
44	NI	33.069	1.19
45	Epi-α-Cadinol	33.185	4.34
46	α-Muurolol	33.352	0.50
47	β-Eudesmol	33.449	0.51
48	α-Cadinol	33.578	1.02
49	NI	33.881	0.45
50	NI	35.751	0.15
Total identified			97.2

RT (min) = Retention times of the compounds; % GC-MS. = Relative percentage of the compound in the EOCC; NI = Not identified.

**Table 3 pharmaceuticals-16-00055-t003:** MIC/MBC or MFC of *C. conduplicatus* essential oil.

Microrganisms	MIC/MBC or MFC (µg mL^−1^)				
	EOCC	AMP	OXA	POL	ANF
*S. aureus* ATCC 25923 (MSSA)	256/512	2/4	2/4	-	-
*S. aureus* ATCC 33591 (MRSA)	512/1024	16/32	32/64	-	-
*E. coli* ATCC 25922	na	nd	-	-	-
*P. aeruginosa* ATCC 27853	na	-	-	1/1	-
*C. albicans* ATCC 10231	na	-	-	-	0.5/1

ATCC—American Type Culture Collection; MIC—minimum inhibitory concentration; MBC—minimum bactericidal concentration; MFC—minimum fungal concentration; EOCC—essential oil from *Croton conduplicatus*; AMP—ampicillin; OXA—oxacillin; POL—polimixin B; ANF—anfotericin B; MSSA—methicillin-sensitive *Staphylococcus aureus*; MRSA—methicillin-resistant *Staphylococcus aureus*; na—no activity; nd—not determined.

**Table 4 pharmaceuticals-16-00055-t004:** Combination testing of *C. conduplicatus* essential oil with antimicrobials agents against MSSA and MRSA strains.

*S. aureus*	Combination	Individual MIC (µg mL^−1^)	Combined MIC (µg mL^−1^)	Individual FIC	FIC Index (FICi)	MIC Reduction (%)	Combination Effect
*S. aureus* ATCC 25923 (MSSA)	OXA/EOCC	2/256	0.5/16	0.25/0.0625	0.3125	75.0/93.75	Synergistic
	AMP/EOCC	2/256	0.25/16	0.125/0.0625	0.1875	87.5/93.75	Synergistic
*S. aureus* ATCC 33591 (MRSA)	OXA/EOCC	32/512	1/32	0.0313/0.0625	0.0938	96.9/93.75	Synergistic
	AMP/EOCC	16/512	0.5/32	0.0313/0.0625	0.0938	96.9/93.75	Synergistic

ATCC—American Type Culture Collection; MSSA—methicillin-sensitive *Staphylococcus aureus*; MRSA—methicillin-resistant *Staphylococcus aureus*; OXA—oxacillin; AMP—ampicillin; MIC—minimum inhibitory concentration; FIC—fractional inhibitory concentration.

## Data Availability

Data are contained within the article.

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
