# Peer review of "Synergistic and Antibiofilm Effects of the Essential Oil from *Croton conduplicatus* (Euphorbiaceae) against Methicillin-Resistant *Staphylococcus aureus"

_pharmaceuticals, 2022, doi:10.3390/ph16010055_

Round 1
Reviewer 1 Report
In this manuscript, authors first extracted and identified the chemical components of the essential oil from Croton conduplicatus, then evaluated their antimicrobial activity and antibiofilm potential. This manuscript represented a complete story; however, the study is insufficient in the data and outcome presentation. Authors must address the following concerns before this can be considered for publication.
1) the data should be revised in a more scientific manner, like concentration for MIC (table 2, 3).
2) the antimicrobial activity and antibiofilm potential of EOCC should be supplemented with experimental images, including agar diffusion method, and colorimetric microdilution method
3) authors must check the grammar errors and polish the language.
Author Response
Dear reviewer, we accept all the suggestions that were made.
1) the data should be revised in a more scientific manner, like concentration for MIC (table 2, 3). THE ITEM WAS MODIFIED AS SUGGESTED
2) the antimicrobial activity and antibiofilm potential of EOCC should be supplemented with experimental images, including agar diffusion method, and colorimetric microdilution method
ITEMS WERE INCLUDED AS SUGGESTED
3) authors must check the grammar errors and polish the language.
An extensive English review was carried out by a specialized company
Reviewer 2 Report
The authors dealt with the antimicrobial and anti-biofilm activities of the essential oils of
Croton conduplicatus alongside the identification of the essential oil components. Comments are listed below.
Language: An extensive language editing is recommended for the whole manuscript as there are too many mistakes to enumerate starting from the abstract.
Abstract: There should be an introductory sentence highlighting a background for the study.
Introduction:
1- The word gram must be Gram, please apply to the whole text.
2- Please use either essential oil or volatile oil , not both.
3- Lines 57-58: That is not true, terpenes could be alcohol, ketones, aldehydes or other classes. Please correct and rephrase.
Results:
1- Lines 81, 108: should be (of) instead of (from).
2- The GC-MS total ion chromatogram must be added as a figure and please include the structures of the major structures.
3- Line 94: major instead of majority.
4- Line 112: MIC was?
5- Table2: write the strains in abbreviated form.
6- Did the author tried using MRSA clinical isolates?
7- Figures 1,2: please improve the resolution.
8- Was there any microscopical photos for the biofilms before and after treatment with the extract and antibiotics?
Discussion: Please highlight the differences and additions of this study compared to the previous studies on the same plant for the MRSA strains in more detail than is already present.
Materials and methods:
1- Please mention the media names in full and give the source of all the chemicals and apparatus used in the study.
Funding: Please add the grant number for the fund.
References:
Please revise to meet the journal style. Journals’ names should be abbreviated, and the page numbers (first-last) must be added. The title of the articles not capitalized.
Author Response
Dear reviewer, we accept all the suggestions that were made.
Language: An extensive language editing is recommended for the whole manuscript as there are too many mistakes to enumerate starting from the abstract. - An extensive English review was carried out by a specialized company
Abstract: There should be an introductory sentence highlighting a background for the study. - We put a short introduction in the abstract within the context of the article
Introduction:
1- The word gram must be Gram, please apply to the whole text – THE ITEM WAS MODIFIED AS SUGGESTED
2- Please use either essential oil or volatile oil , not both. - THE ITEM WAS MODIFIED AS SUGGESTED: The expression volatile oil was modified to essential oil.
3- Lines 57-58: That is not true, terpenes could be alcohol, ketones, aldehydes or other classes. Please correct and rephrase. - The item has been fixed
Results:
1- Lines 81, 108: should be (of) instead of (from). - The ITEM WAS MODIFIED AS SUGGESTED
2- The GC-MS total ion chromatogram must be added as a figure and please include the structures of the major structures. – It was included as requested
3- Line 94: major instead of majority. - ITEM WAS MODIFIED AS SUGGESTED
4- Line 112: MIC was? THE PHRASE WAS MODIFIED TO BETTER UNDERSTAND: because the MIC values against these strains were > 1024 µg mL-1
5- Table 2: write the strains in abbreviated form. - ITEM WAS MODIFIED AS SUGGESTED
6- Did the author tried using MRSA clinical isolates? It was not used in clinical isolates in this study, the principal aim was to verify the antibacterial activity of Croton conduplicatus essential oil against standard strains. The study against MRSA clinical strains may be development in future, using a larger number of strains, since it was observed effective activity of Croton conduplicatus essential oil against MRSA in planctonic cells, biofilm formation and formed biofilm with synergistic effect.
7- Figures 1,2: please improve the resolution - The figures quality were improved.
8- Was there any microscopical photos for the biofilms before and after treatment with the extract and antibiotics? Microscopic studies with the biofilm were not performed in this study, only the quantification of biomass, where a reduction in biomass was observed with the treatments when compared to the biomass of the positive control (no treatment). And future studies with MRSA clinical strains are of interest to perform tests such as scanning microscopy on these treated biofilms. Photos from the biomass reduction experiment were included to illustrate the antibiofilm activity.
Discussion: Please highlight the differences and additions of this study compared to the previous studies on the same plant for the MRSA strains in more detail than is already present.
We have included a paragraph showing the antibacterial activity of some species of the genus Croton with chemical composition similar to that of Croton conduplicatus
Materials and methods:
- Please mention the media names in full and give the source of all the chemicals and apparatus used in the study. - ITEMS WERE MODIFIED AS SUGGESTED
- Funding: Please add the grant number for the fund. OK
- References:
Please revise to meet the journal style. Journals’ names should be abbreviated, and the page numbers (first-last) must be added. The title of the articles not capitalized. The references were corrected.
Reviewer 3 Report
The manuscript entitled “Synergistic and antibiofilm effects of the essential oil from Croton conduplicatus (Euphorbiaceae) against methicillin resistant Staphylococcus aureus” is an important scientific paper focuses mainly on the extraction of essential oil from Croton conduplicatus and its analysis by GC-MS. The essential oil and its synergistic effect with antibiotics were also included for antimicrobial activity. The manuscript is accepted with the following comments.
1. Materials and methods: Include the figure of Clevenger apparatus with pant materials in round bottom flask.
2. Results: Include GC-MS chromatogram of essential oil of Croton conduplicatus.
3. Conclusions: The effectiveness (antibacterial activity) of the essential oil of Croton conduplicatus is quite impressive. Compile a Table and compare your results with other essential oils from the same plant family or others. The new findings of the compounds and a decrease or increase in % of major compounds can be included.
Author Response
Dear reviewer, we accept all the suggestions that were made.
1 Material and methods: Include the figure of Clevenger apparatus with pant materials in round bottom flask.
Unfortunately we were not able to get this picture. Instead, we have added a picture with the yellowish essential oil from Croton conduplicatus flasks where it was stored under refrigeration.
- Results: Include GC-MS chromatogram of essential oil of Croton conduplicatus. It was included as requested.
- Conclusions: The effectiveness (antibacterial activity) of the essential oil of Croton conduplicatus is quite impressive. Compile a Table and compare your results with other essential oils from the same plant family or others. The new findings of the compounds and a decrease or increase in % of major compounds can be included.
The table has been added
Round 2
Reviewer 1 Report
I have no more comments on this revised manuscript.
Authors have addressed all my comments and improved the quality of this manuscript.
Reviewer 2 Report
The authors have addressed my comments.